# Processing Photoplethysmograms Recorded by Smartwatches to Improve the Quality of Derived Pulse Rate Variability

**DOI:** 10.3390/s22187047

**Published:** 2022-09-17

**Authors:** Adam G. Polak, Bartłomiej Klich, Stanisław Saganowski, Monika A. Prucnal, Przemysław Kazienko

**Affiliations:** 1Department of Electronic and Photonic Metrology, Wrocław University of Science and Technology, 50-317 Wrocław, Poland; 2Department of Artificial Intelligence, Wrocław University of Science and Technology, 50-370 Wrocław, Poland

**Keywords:** PPG, ECG, PRV, HRV, artifact reduction, wearables, biomedical signal processing

## Abstract

Cardiac monitoring based on wearable photoplethysmography (PPG) is widespread because of its usability and low cost. Unfortunately, PPG is negatively affected by various types of disruptions, which could introduce errors to the algorithm that extracts pulse rate variability (PRV). This study aims to identify the nature of such artifacts caused by various types of factors under the conditions of precisely planned experiments. We also propose methods for their reduction based solely on the PPG signal while preserving the frequency content of PRV. The accuracy of PRV derived from PPG was compared to heart rate variability (HRV) derived from the accompanying ECG. The results indicate that filtering PPG signals using the discrete wavelet transform and its inverse (DWT/IDWT) is suitable for removing slow components and high-frequency noise. Moreover, the main benefit of amplitude demodulation is better preparation of the PPG to determine the duration of pulse cycles and reduce the impact of some other artifacts. Post-processing applied to HRV and PRV indicates that the correction of outliers based on local statistical measures of signals and the autoregressive (AR) model is only important when the PPG is of low quality and has no effect under good signal quality. The main conclusion is that the DWT/IDWT, followed by amplitude demodulation, enables the proper preparation of the PPG signal for the subsequent use of PRV extraction algorithms, particularly at rest. However, post-processing in the proposed form should be applied more in the situations of observed strong artifacts than in motionless laboratory experiments.

## 1. Introduction

For many years, electrocardiography (ECG) has been considered the gold standard in the field of cardiac testing. It is an essential tool for diagnosing various cardiac diseases. It is also commonly applied to derive another signal—heart rate variability (HRV), whose high diagnostic ability has been demonstrated in recent years [1,2,3]. In particular, the analysis of frequency components, including high (0.15–0.4 Hz), low (0.04–0.15 Hz), very low (0.004–0.04 Hz), and ultra-low (<0.004 Hz) bands, is of actual interest, as they reflect, among others, sympathetic and the parasympathetic activity, blood pressure, thermoregulation, renin–angiotensin mechanism, or circadian rhythms [4]. ECG requires the usage of electrodes attached to the body, which is not considered as convenient, especially when moving. One of the latest developments is the use of a chest strap with measuring electrodes [5,6]. However, this solution still causes inconveniences in everyday life, e.g., squeezed chest feeling and possible skin rash. Recently, cardiac monitoring based on photoplethysmography (PPG) is gaining more and more popularity also due to its lower cost and greater usability. It is widely used in various wearable devices, such as smartwatches and wristbands. The ability to measure someone’s PPG may be used to monitor her/his affective state, sleep quality, and other aspects impacting overall well-being [7,8,9,10,11]. Unfortunately, PPG is negatively affected by various types of disruptions related to device shifting around, changes in lighting, or the intensity and nature of body movements. In particular, these artifacts interfere with the algorithms that extract pulse rate variability (PRV) from the PPG, introducing serious errors into this diagnostically valuable signal.

The above-mentioned noticeable interest in the PRV signal acquisition by wearable devices and its reliable processing to extract encoded information has prompted many research groups to sophisticated work on reducing artifacts in PPG. The vast majority of these studies has been summarized in recent review articles [12,13,14,15,16,17,18]. Overall, the approaches used fall into three general groups. The first of them includes signal decomposition methods using the fact that some artifacts are located in specific components and as such can be removed from the signal reconstructed from the other ones. Among them, the most popular are: discrete wavelet transform (DWT), empirical mode decomposition (EMD) or independent component analysis (ICA), and less frequently applied: variational mode decomposition (VMD), short-time Fourier transform (STFT) or singular value decomposition (SVD). Their common advantage is the ability of removing artifacts without using any other signal recorded simultaneously. The second group covers adaptive filters that require an additional signal, and these are sensitive to the sources of artifacts included in the PPG. An inherent feature of such filters is the removal of components represented by the accompanying data from the signal under consideration. Since usually most harmful artifacts come from body or arm movements, the signal preferred here is from an acceleration sensor built into same wearable. The most frequently used adaptive filters are: least mean squares (LMS, both in linear and nonlinear version), recursive least squares (RLS) and the Wiener filter. The third general set consists of other methods of a different nature. Among them, one can find classical spectral analysis (for artifacts characterized by a frequency content diverse from the PPG spectrum), methods examining the statistical parameters of signal samples, or the most recently developing approach, i.e., machine learning methods. It is also worth noting the trend of combining multiple of the above-mentioned types of methods, which often demonstrate complementary properties.

The aim of this study is to identify the nature of artifacts caused by various types of factors under the conditions of precisely planned experiments as well as to propose and validate methods for artefact reduction based solely on the PPG signal. Their goal is to improve the accuracy of PRV derivation, paying simultaneously special attention to preserving the frequency content of PRV.

A big role in the selection and effectiveness of artifact removal methods is knowledge about linking their character with specific conditions. For this reason, in many studies, the PPG signal was recorded during experiments inducing artifacts. Most often, they included: (1) baseline measurements [19,20,21,22,23]; (2) controlled movements of an arm or fingers (please note that the experiments with the arm and finger movements are compatible because the choice of the type depended on the position of the PPG sensor on a wrist or finger) [20,24,25,26,27,28,29,30,31,32,33], (3) breathing with different patterns [22,34], (4) walking or running on a treadmill with a regulated pace [21,24,25,35,36], or (5) tapping the sensor [26]. In many studies, the correctness of PPG processing was assessed by comparing it with the results of ECG processing, which was taken as the reference data. Since one of the effects of artifacts is the different number of detected heartbeats in the synchronized ECG and PPG signals, a direct comparison of both derived HRV and PRV signals is not trivial. Rather, to avoid this problem, different averaged metrics were used for the assumed time intervals, such as the heart rate (HR) or statistical measures, so that the number of samples in both signals was the same [34,36,37,38,39,40,41,42,43,44,45,46]. Direct comparisons between PRV and HRV extracted from the PPG and ECG have been studied only a few times. In [47,48], the corrupted fragments of PPG and the associated R-R intervals from the ECG were ignored, so that the rest of the pulses could be unambiguously matched. The root mean square of subsequent differences was calculated by Lam et al. but without any explanation of how the problem with a different number of samples was resolved [49]. Another proposed approach was to interpolate these time series with a constant sampling rate using splines [22]. Nevertheless, more complete approaches to directly comparing HRV and PRV signals are still needed.

The main contribution of this study is to propose a series of methods for the reduction of artifacts affecting PPG based solely on this signal and focusing on the proper derivation of PRV, thus maintaining its frequency content, which is crucial for subsequent PRV analyses. The methods were tested and selected on the basis of carefully designed experiments with an increasing level of artifacts. In particular, the advantage of PPG amplitude demodulation in determining the duration of pulse cycles was demonstrated as well as the benefits of PRV post-processing, including statistical correction of outliers and prediction of PRV samples using the autoregressive (AR) model. We also demonstrated the possibility of quantifying the correctness of the extracted PRV against the reference HRV derived from the simultaneous ECG using dynamic time warping for data of unequal length or classical measures for the uniformly resampled signals.

The organization of the paper is as follows. The first part of Section 2 provides a description of data acquisition with a detailed protocol of experiments, while the second part contains a description of the methods used to process the ECG and PPG signals. The end of this section is a presentation of the post-processing algorithms and the evaluation procedure and metrics. Section 3 shows the results obtained by the applied algorithms on the PPG signal with reference to the ECG. The discussion of the results, taking into account also limitations and problems encountered in this work, is included in Section 4. Finally, Section 5 summarizes the work.

## 2. Materials and Methods

### 2.1. Experimental Protocol and Collected Data

The study was conducted in accordance with the Declaration of Helsinki and ethical guidelines provided by the National Science Centre in Poland. All participants were informed about the study before, and all provided signed written consent.

We collected data from 11 participants (three females) aged between 22 and 34 (mean = 25, SD = 3.7). All participants were students or university employees. Only one person reported being diagnosed with left ventricular hypertrophy in the past. Other participants did not report any heart-related problems.

We exploited three wearable devices simultaneously worn by all participants: (1) Samsung Galaxy Watch 3, (2) Polar H10, and (3) Empatica E4. The smartwatch Samsung Galaxy Watch 3 was placed on a non-dominant hand to gather the photoplethysmograph—PPG (with 25 Hz sampling rate) and accelerometer—ACC (50 Hz) signals. The chest strap Polar H10 was utilized to obtain the reference signals: electrocardiogram—ECG (130 Hz) and ACC (200 Hz). The E4 Manager desktop application was used to download the data stored in the Empatica E4, while for the acquisition of signals from the Samsung and Polar, we had developed our own mobile application working in real time with Bluetooth transfer. The Polar H4 appears to be a good and comfortable alternative to the Holter monitors when performing intense activities [6,50]. Additionally, we placed the wristband Empatica E4 on the dominant hand to acquire PPG (32 Hz) and ACC (64 Hz) signals. However, data from the Empatica are not considered in this article, because it is equipped with embedded pre-processing routines and returns the PPG signals devoid of some artifacts, especially of low-frequency content. Therefore, in this paper, we focused on the more corrupted signals recorded by Galaxy Watch 3 to make the study more complete, leaving the data from the Empatica for later use.

The experiments were divided into three sessions (Table 1 and Figure 1): (1) a laboratory session, (2) a running session, (3) and an everyday life session. The sessions were intended to contain an increasing level of motion artifacts. Before each session, the participants were instructed about the experiments and how to prepare and use the devices (e.g., to moisten the chest strap’s electrodes). Before and after each session, the participants simultaneously tapped all three devices together to enable synchronization of the signals across the different devices.

The laboratory session consisted of the following five activities:***NoMove.*** Participants were sitting still in a comfortable position on a chair, with hands resting on a table, normally breathing, and without talking. The room lighting (a lamp) was constant. With this setup, we intended to have the minimal number of artifacts in the signal.***Light***. Participants were sitting still as in the *NoMove* experiment. The room lighting was being switched on and off every ten seconds (using a metronome to measure the interval).***Tap***. While in a sitting position, the participants were tapping the Samsung Watch 3 with a finger, imitating a typical smartwatch usage. The rhythm of the tapping was determined with the metronome, i.e., with one-second periods during the first 150 s of the experiment and two-seconds periods during the remaining 150 s.***Arm***. While sitting, the participants were constantly raising and putting down a straight arm, possibly in one plane. The movement pace was guided with a sine wave displayed on a monitor—ten seconds for each direction of movement. The goal of this experiment was to analyze, i.a., the effect of changes in hydrostatic blood pressure on artifacts.***Breath***. While sitting, the participants were performing a controlled breathing, which was guided with a sine wave, i.e., five breaths per minute during the first 150 s of the experiment (very slow and deep breath), and 30 breaths per minute during the remaining 150 s (fast breathing).

Each experiment lasted five minutes. The interval between experiments was 20 s, during which the participant should not move but relax. The entire laboratory session lasted about 30 min.

The running session included three experiments, each with different pace, i.e., 1, 5, and 9 km/h; equivalent to about 40, 110, and 150 steps per minute, respectively. Each activity lasted five minutes, and the interval between them was one minute. The pace was selected based on the assumption that the lower and higher number of steps per minute will be significantly different from the recorded HR value, whereas the middle value will be similar to the observed HR value.

The everyday life session covered at least seven hours of participants’ regular day as well as a minimum of five hours of their sleep.

### 2.2. General Methodology

This paper focuses on analyzing the PPG and ECG signals from the laboratory session only, with the data from the treadmill and everyday life sessions postponed for use in further research. This is related to the goal of this study, which is the use of artifact reduction methods based solely on the PPG signal. Meanwhile, removing artifacts from the treadmill and everyday life sessions would require taking into account the ACC signals as well, since then, the PPGs contain even stronger motion artifacts than in the analyzed “dynamic” experiments.

Processing of the PPG and ECG signals for PRV and HRV extraction has been divided into the following stages (Figure 2): pre-processing including signal synchronization, HRV and PRV derivation, post-processing, and evaluation by comparing HRV and PRV obtained from the ECG and PPG analyses. The methods used at each stage of the analysis for the PPG and ECG signals are different in some cases, as shown in Figure 2 and detailed in Section 2.3 and Section 2.4. Subsequently, to obtain the highest possible accuracy of PRV from the PPG signal, it was decided to compare three popular methods of its extraction (see Section 2.4). All signal processing procedures were implemented in Matlab (R2022a, MathWorks, Natick, MA, USA).

### 2.3. ECG Processing

In general, the ECG obtained from the Polar H10 device was of a good quality. To reduce high-frequency noise and baseline wander, occurring especially at the beginning of the measurements, we applied the discrete wavelet transform (DWT) [51] but in a form with maximal overlap, which does not change the original length of the signal. The wavelets allow transforming a time-value signal into the time-scale components. The similarity of the signal with the wavelets is calculated separately with a sliding time window. The use of the DWT is often referred to as a cascaded band-pass filter and called wavelet decomposition. It consists of decomposing a given signal into a specific series of coefficients with decreasing frequencies, using a bank of filters.

The *Symlet 4* wavelet was chosen to decompose the ECG signal because of its similarity to the QRS complex [52]. We used ten decomposition levels so that the approximation coefficients covered frequencies below 0.064 Hz. To obtain the filtered signal, the first two detailed coefficients (between 16.25 and 65 Hz) and the approximation coefficients were rejected, as they contained the most signal disruption. Next, the signal was reconstructed from the remaining coefficients, using the inverse DWT (IDWT).

To extract HRV from the ECG signal, we utilized the Pan–Tompkins algorithm [53]. Originally in the algorithm, prior to the determination of the QRS complexes, signal processing is applied to remove noise and to expose the feature under study. In our case, band-pass filtering composed of cascaded low-pass and high-pass filters was not needed, because we had already applied the wavelet transform (which is more effective at removing noise).

### 2.4. PPG Processing

After analyzing various wavelets used for PPG denoising [54], we started the filtering of the PPG signal by applying DWT with the *Coiflets 1* as the base wavelet, since it suits well the PPG pulse wave consisting of systolic and diastolic phases. Because the main useful frequency range of the PPG signal is 0.8 to 2.5 Hz [55] and the sampling rate was 25 Hz, four levels of decomposition were performed. As a result, DWT provided the 3rd and 4th detail coefficients associated with frequencies 0.78–1.56 Hz and 1.56–3.12 Hz, respectively (the total limits correspond to the HR from 47 to 187 bpm), from which the filtered PPG signal was reconstructed.

The next step in processing the PPG signal was amplitude demodulation. We analyzed three algorithms reducing amplitude fluctuations, i.e.,

**Adaptive standardization,** which consists of (1) calculating standard deviation over a sliding window of the odd length and (2) dividing by the obtained deviation the signal value of the middle sample in the considered window.**Online algorithm** performed with the following steps: (1) selecting a window of odd length; (2) subtracting from the signal the constant component; (3) calculating the difference between global maximum and minimum, further referred to as *Delta*; (4) determining local maxima and minima with the following conditions: max>P×Delta or min<−P×Delta, where *P* is an adjustable parameter; (5) obtaining the average of all local maxima and minima; (6) dividing by the obtained average the middle sample in the considered window; (7) sliding the window by one sample and repeating steps 2–7 until the window reaches the last sample in the analyzed signal.**Envelope-based** filtering. We considered several approaches to determine envelopes, i.e.,:(a)Using spline interpolation over local maxima separated by a given number of samples;(b)Calculating root mean square of the signal;(c)Using the magnitude of the signal, which is computed by filtering with a Hilbert-FIR filter over the sliding window;(d)Applying discrete Fourier transform implemented as the Hilbert one, which returns symmetric envelopes used for signal demodulation by dividing the samples by the local envelope value, and it is parameter-free.

Finally, we decided to use the last one 3(d), because it provided satisfactory results.

To derive PRVs from the PPG signals, three algorithms were tested:**PDA**—Peak Detection Algorithm [56];**AMPD**—Automatic at Multiscale-based Peak Detection [57];**SSF**—Slope Sum Function [58].

The PDA is a simple algorithm that compares neighboring samples to find local minima and maxima. We used a peak detection threshold set to 0.8.

The AMPD has been designed and is commonly used for noisy periodic and quasi-periodic signals. It is based on the calculation of the local maxima scalogram, which is computationally expensive. To apply it to a longer signal, e.g., 5 min or longer, we had to divide the PPG signal into shorter parts, i.e., 30 peaks, which in our case was about 20 s.

The SSF is widely used to determine systolic peaks in a PPG signal obtained from a wrist or finger [58,59,60]. It applies a transformation function defined as: (1)$$\mathrm{SSF}=\sum_{k=i-w}^{i}\Delta x_{k}\;\mathrm{where}\;\Delta x_{k}=\left\{\begin{array}{cc} \Delta s: & \Delta s>0\\ 0: & \Delta s\leq 0 \end{array}\right.$$
where w=128 ms is the length of the considered window, and *s* is the length of the PPG signal.

The decision rule was established by a threshold calculated as the mean of the first ten seconds of the signal, which is updated by 40% of the maximum value of SSF transformation for each detected pulse. The SSF algorithm determines each pulse onset of the signal, which is the dicrotic point, so in order to obtain systolic peaks, the analyzed signal had to be reflected upside-down.

### 2.5. Signal Synchronization

Since the ECG and PPG signals were collected with two different devices, it was necessary to develop a synchronization procedure, which was applied after DTW/IDTW filtering for both PPG and ECG signals. For that reason, each session started and ended with tapping the devices against each other. To observe the taps in the ACC signal better, it was sequentially processed. The squares of the three-axis signals were summed, and then, the arithmetic mean was subtracted. Finally, to smooth the signal, the moving average with a 20-, 34-, and 16-sample window length (experimentally chosen for the Samsung, Polar, and Empatica devices, respectively) was applied.

Next, the Hilbert transform was used to obtain the envelope, which made the tapping easier to see. Eventually, the signals were normalized to the range [0, 1]. The timestamps corresponding to the taps were manually obtained from such ACC signals. Since the Polar H10 does not provide timestamps for all samples, the timestamps from the Samsung Watch 3 were used to create a time vector for a particular session, and the samples within a given session were distributed equally across the time vector. Both signals, ECG and PPG, were then upsampled by interpolation with cubic splines to 1 kHz to enhance the HRV and PRV quantization levels, standardize, and facilitate the analysis between the signals (Figure 2).

The alignment of the ECG and PPG signals was tuned manually by comparing and adjusting their waveforms, i.e., R-waves and systolic peaks.

### 2.6. Post-Processing

After computing the HRV and PRV waveforms, post-processing was applied to possibly increase their reliability, taking into account both a priori information about HRVs series and their local properties (Figure 2). First, a wide but limited physiological HR range was considered. For young athletes, the resting HR can be as low as 44 ± 2 bpm [61], and the maximum pulse, depending on age, can be estimated for people aged around 20 years as approximately 198 bpm [62]. Therefore, the expected physiological R-R or pulse-to-pulse intervals must be 0.3–1.5 ms (200–40 bpm, respectively). Taking this into account, both PRV and HRV were screened [49]. If the interval was less than 0.3 ms, the second detected peak was treated as an artifact, and this interval was combined with the next one. Alternatively, intervals longer than 1.5 ms, suggesting that the actual pulse had been skipped, were split into half values. Moreover, since PRV were particularly strongly influenced by artifacts, they were additionally fixed using the moving average of 25 samples (the value corresponding to the order of the AR model used in the next step) and its standard deviation (SD). In particular, when a given sample was outside the range of mean ± 3 × SD (the “three-sigma rule” ensuring that the probability of correcting a valid value is less than 0.003), it was processed in the same way as previously. The next post-processing stage involved resampling the HRV and PRV data unevenly distributed over time by cubic spline interpolation. It is necessary to allow both direct comparisons of ECG and PPG derived signals (the same number of evenly spaced samples) and the limitation of the HRV and PRV spectra to the physiological bound of 0.5 Hz by digital filtering. For that purpose, both HRV and PRV series were first oversampled to 10 Hz, then low-pass filtered to 0.5 Hz, and finally downsampled again to 1 Hz. Having PRV sampled at this constant frequency, it was also possible to correct their local frequency properties using autoregressive modeling, since AR model coefficients cover spectral information [63]. In such an approach, a 25-sample window [63] was slid across the PRV data to estimate AR model local coefficients and to predict the next sample together with its SD. If the relevant PRV sample differed from the predicted value by more than the SD, it was replaced with that prediction.

### 2.7. Evaluation Procedure and Metrics

Complete processing of the ECG and PPG signals returned HRVs and PRVs for the three methods of PPG processing after several stages: raw signals, HRVs and PRVs fixed according to physiological and statistical premises, data uniformly resampled and low-pass filtered, as well as PRVs with locally corrected spectra. Appropriate metrics should have been applied to assess the impact of the subsequent signal processing stages on the overall quality of final PRV. It is possible to compare two associated time series with a different number of samples using a dynamic time warping (DTW) technique that aligns the corresponding signals [64,65]. Thanks to this, the DTW-defined distances between these signals were calculated before and after repairing the HRV and PRV series. Linear correlation coefficients (*r*) for HRVs after their aligning were also computed. The next comparisons were made for the evenly sampled signals (before and after autoregressive correction) by calculating the relative root mean squared errors (RRMSE): (2)$$\mathrm{RRMSE}=\sqrt{\sum_{i=1}^{n}\frac{{\left({\mathrm{PRV}}_{i}-{\mathrm{HRV}}_{i}\right)}^{2}}{{\mathrm{HRV}}_{i}^{2}}}\times 100\%\;$$
where *n* is the length of the HRVs vectors and *i* is the element index. Then, these results were averaged (mRRMSE) across the study participants for a particular experiment: (3)$$\mathrm{mRRMSE}=\sqrt{\sum_{i=1}^{N}\frac{\mathrm{RRMSE}{\left(P_{i}\right)}^{2}}{N}}$$
where *N* is the number of participants for each experiment, and $P_{i}$ is the participant index. For each of the metrics, ordinary means and SD were computed.

Finally, the effect of post-processing on the PRV frequency content was examined by calculating the power spectral density (PSD) of the HRV as well as PRV signals—before and after pre-processing, and then comparing them. PSD was estimated for 5 min waveforms of data sampled at 1 Hz using the Welch method with a window length of 30 s and 25 samples overlapped.

## 3. Experimental Results

The raw PPG signals collected by the Samsung Watch 3 on Person 6 (P6) during the further analyzed experiments are shown in Figure 3. In addition, an exemplary raw PPG signal characterized by a significant baseline wandering is depicted in Figure 4a. This 10 s fragment was selected as well representing the effects of subsequent steps of processing the data from the *Arm* experiments, which were characterized by hard-to-remove artifacts. By applying the DWT/IDWT to that signal, we managed to decrease the wandering effect, as shown in Figure 4b. Removing the amplitude modulation helped us to further improve and standardize the signal, as shown in Figure 4c. This, in turn, enabled us to apply the PRV extraction algorithms. The results of applying the AMPD algorithm are shown in Figure 4d.

On the other hand, the ECG signal obtained with the Polar H10 device presented satisfactory quality, even when participants were performing intense body movement. To reduce the baseline wander and high-frequency content of the signal without affecting the QRS complex, we applied the DWT/IDWT denoising method. Figure 5 contains an example of the filtered ECG signal with peaks detected by the Pan–Tompkins algorithm as well as the filtered PPG signal with peaks detected by the AMPD, PDA, and SSF algorithms. The time shift between the R peaks in ECG and systolic peaks in PPG, caused by the pulse transit time, is roughly constant throughout the whole experiment. Moreover, the systolic peaks are located right behind the T wave, which indicates the signals were appropriately synchronized and the sampling frequencies were correctly calculated.

The comparison of HRV obtained from the ECG signal (with the Pan–Tompkins algorithm) and PRVs from the PPG signal (with the AMPD, PDA, SSF algorithms) before and after signal processing is presented in Figure 6. It can be noted that filtering removed outliers in both ECG and PPG signals and improved the correlation between HRV and PRV obtained from two source signals. It is also visible in Figure 7 that PRVs post-processing improved their frequency content compared to the HRV spectra.

More detailed results are available in Table 2. It can be seen that for “static” experiments (*NoMove* and *Light*), PRVs obtained from the PPGs are very similar to HRVs obtained from the ECGs even without additional signal processing—similarity measures (RRMSE and DTW) have values lower than 3.6% and 4.9, respectively, and the Pearson correlation coefficient is close to 1. For experiments with movements (*Tap* and *Arm*), the PRV and HRV signals derived from the PPG and the ECG differ significantly. The proposed signal filtering helped to obtain more proper PRV from the PPG in such a case, although the Pearson correlation coefficient was still at the 0.62–0.77 level. For the *Breath* experiment, the similarity of the PRV and HRV from the PPG and ECG is rather satisfactory, achieving *r* > 0.85.

We noticed that for some participants, the HRVs similarity measures were significantly lower in case of a few experiments, i.e., P8 (participant no. 8) and P9 in the *Breath* experiment, P7 and P8 in the *Arm* experiment, and P5, P8, P9, and P10 in the *Tap* experiment. Since there are only a few cases like this, and the majority of them are related to the experiments introducing movement, we conclude these are outliers caused by insufficient adhesion of the sensor to the skin (loosely worn device). Table 3 contains the HRV and PRV similarity measures with the outliers removed. Compared to Table 2, it can be seen that the obtained results for all “dynamic” experiments demonstrate a clear improvement, particularly for the *Breath* activity (reduction of the RRMSE by about 75%). Additionally, we performed a statistical paired Student’s t-test with the 5% significance level for the PRV data to check for differences in the performance of the PRV extraction algorithms. It stems that the SSF indicates more errors in the *Tap* and *Arm* experiments than other algorithms.

To determine the computational complexity, we measured the execution time of each PRV algorithm applied to the same signal fragment lasting five minutes. The computation was repeated ten times. On average, the AMPD required 20.6 s to process the signal fragment and provide peaks, whereas it took only 0.08 s for the SSF and 0.01 for the PDA algorithm to compute the results.

## 4. Discussion

### 4.1. Novelty of This Study

The primary aim of this study was to propose a procedure for reducing artifacts caused by various factors induced during the planned experiments based solely on the PPG signal. Particular attention was paid to improving the accuracy of PRV derived from the PPG compared to HRV derived from the concurrent ECG. It referred to maintaining its frequency content, as this is the main feature used in subsequent analyses. The PSDs presented in Figure 7 show similar frequency characteristics of both HRV and PPG, especially in the case of experiments with less artifacts, lower frequencies, and after post-processing. Our methodology is in contrast to the most popular approaches [12,15,24,33,35,36,37,39,42,66], since it does not require the use of any sensor other than the PPG.

Experiments included in this study were of various types, some of them replicating the ones in papers mentioned in the Introduction. We focused our research on their potentially progressive impact on PPG quality to successively deal with new artifacts. Most of the articles published until now have included baseline measurements and controlled arm movements, typically lasting 1 min or less [21,24,25,26,36], or 5 min [19,20,22,24,34,67,67], and rarely longer than 5 min [23,35]. The advantage of our study was that each activity lasted at least 5 min, and sessions of 5 and 7 h of continuous monitoring were also included. Very often, experiment protocols in the more complex cases described in the literature contained at least 1 min of rest to restore a normal heart rate. Due to the arrangement of the experiments according to their intensity and their division into three sessions, the breaks in the laboratory sessions were shortened to 20 s, but in the complex experiments (running and everyday life session), they remained 1 min.

In studies carried out by others, the ambient light was not changed, especially cyclically, compared to our experiments conducted as *Light*. The exception is [40], where changes in the light entering the detector were examined with the 3D-printed rings for the smartwatches with and without holes to change the amount of light falling on the sensor, and this caused a deterioration in the quality of the reading. In research on breathing artifacts [22,34], the methods for monitoring respiratory rhythm were not provided, while in this study, a sine wave of a specific frequency was displayed, which makes it easy to repeat this protocol by others in the future.

The decision to temporarily set aside data related to the treadmill and everyday life in this first approach to the analysis of the collected experimental results is associated with the observed relationship of the induced artifacts to the rhythm of steps and their overlapping with the heartbeat, so that their reduction without an additional signal such as ACC (also recorded during the experiments) seems to be unlikely. On the other hand, due to the measurements of PPG on different parts of the body presented in the literature [68], differences in possible artifacts should be taken into account. The finger PPG is much less susceptible to motion artifacts than the wrist due to the more stationary nature of the measurement [69].

### 4.2. Justification of the Methods

Some previous publications [6,50] compared the Polar H10 to Holter’s devices, which are recognized as the gold standard in ECG measurements and are still considered the best approach for determining the R-R intervals during basic activities. However, they might be not suitable for high-intensity activities. It was shown in these papers that Polar H10 is comparably accurate with even better quality of the recorded signal at intense activities with strong body movements. While it is typical to use in research the R-R intervals calculated by the H10 device itself, we decided to independently derive HRVs from the collected ECGs (see Section 2.3), as it had been shown that the embedded firmware can return an HRV signal with some uncontrolled errors due to missed or mis-detected R-peaks [6]. For the above reasons, HRVs derived from the ECG signals recorded by Polar H10 were considered an appropriate reference for our study. Both raw ECG and PPG signals were upsampled to 1 Hz by spline interpolation, which theoretically guarantees a 1 ms resolution of derived HRV and PRV values. However, such R-peak or pulse position accuracy cannot actually be expected, because the original sampling rates (130 and 25 Hz, respectively) together with additional filtering by the DWT/IDWT limited the spectra and therefore also modulated the waveform shapes. On the other hand, the sampling rate of 250 Hz (i.e., 4 ms interval) has been shown to be acceptable in HRV frequency-domain analyses [70]; therefore, it can be anticipated that the above-mentioned factors will not cause this 4 ms inaccuracy to be exceeded.

Artifact removal with DWT/IDWT is one of the methods based on signal decomposition, which is often used interchangeably with empirical mode decomposition (EMD) [15]. However, the main advantage of DWT is that the decomposed coefficients correspond to specific frequency ranges in contrast to the intrinsic mode functions obtained from EMD. This makes it easier to choose which coefficients contain unwanted artifacts. The DWT/IDWT works as a more advanced band-pass filter, which has proven to be very efficient at removing the slow components visible in Figure 4a,b as well as high-frequency noise. For a different sampling frequency of the signal, however, the frequency ranges of the individual coefficients also change, which should be borne in mind depending on a specific application.

One of the motion-related artifacts is the amplitude modulation of the PPG, which is undoubtedly a nonlinear process. There are some other papers that have dealt with this problem, for example in order to train a neural network [30] or for noise cancellation and SpO2 estimation [71]. An alternative to determining the envelope by means of the Hilbert transform used in this study is, for instance, computing a local value of the standard deviation or the RMS of the signal [72]. Either way, the main benefit of amplitude demodulation is better preparation of the PPG to determine the duration of pulse cycles as well as reducing the impact of some other artifacts. However, another effect of demodulation is also the loss of useful information contained in amplitude changes and related to physiological processes, such as respiratory rate [73]. However, when using the PPG signal only to determine PRV, it is not essential.

### 4.3. Significance of the Results

Three algorithms were used to derive the PRV waveforms from the PPGs, which differ significantly in the way they determine the duration of pulse cycles. The first difference noticed is the computational effort: the most greedy is the AMPD, then SSF, and the fastest is PDA because of its simplicity. On the other hand, the AMPD algorithm is convenient to use due to the lack of hyper-parameters and no restrictions on the frequency of the tested signal. The most influential factor in the results between used algorithms was the selection of appropriate hyper-parameter values. As shown in Figure 4d and Figure 5, each of the algorithms allowed for the effective determination of the PRV waveforms from the processed PPG signal.

Looking at the post-processing effects, significant improvements can be seen in the resulting PRV quality measures, as shown in Table 2 and Table 3. Post-processing improved the results primarily in the more complex experiments, such as *Tap* and *Arm*, but it sometimes worsened slightly in simple cases, such as *Light*. This is mainly due to the effect of the AR model, because when it operates on a very good quality PPG signal, then sometimes, the correct PRV values are replaced by the prediction outcomes. This leads to the conclusion that post-processing in the proposed form should be applied more in the situations of observed strong artifacts than in motionless laboratory experiments. Nevertheless, it improves the PRV spectra compared to HRV (Figure 3). It is worth adding here that the further optimization of post-processing algorithms’ hyper-parameters is still possible, which may bring even better results. Moreover, it was observed that the best results of applying the AR model were obtained when the beginning of the PRV waveforms were determined without distortions, so that the prediction of subsequent samples was much more accurate and the changed values were more relevant, additionally allowing the algorithm to work properly after sliding the window.

The best mRRMS scores (2.5% in Table 2 and Table 3) were obtained in the “static” measurements during the *Light* experiment, which is interesting because they are better than for the *NoMove* experiment. This indicates that changes in ambient light have not affected the performance of the algorithms. On the other hand, the worst outcomes are for the *Tap* experiments (Table 2 and Table 3). This is mainly because the strong tapping introduced notably erroneous peaks into the PPG signal. This is especially visible when comparing these results with the *NoMove* experiments, as the only difference between them is tapping. Several times higher values of the dissimilarity metrics between HRVs and PRVs presented in the tables indicate a sharp deterioration in the peak detection accuracy caused by tapping, because the PRV signals are constructed directly from the peaks determined by the three tested algorithms. In addition, the intense arm movements affected blood flow in the limb through neural regulation and change in blood pressure [74,75], modulating the amplitude of PPG. Fortunately, the amplitude was successfully demodulated (Figure 4), and other such additive artifacts can be removed by adaptive filters using an additional ACC signal [24,36,66], and moreover, strong tapping in a smartwatch is rare. After the rejection of outlier subjects (Table 3), it can be seen that much better results have been achieved for the *Tap* and *Arm* and especially *Breath* experiments, while the outcomes for the “static” activities remain unchanged. Large discrepancies in results for different people may indicate differences in the strength of the sensor’s attachment to the body, which becomes particularly significant during movements. Therefore, only the results of the “dynamic” experiments have improved after the outliers rejection. An additional conclusion is that a personalization approach to each participant should be considered.

From the statistical tests performed regarding the methods for extracting PRV from PPG (Table 2 and Table 3), the differences on the 5% significance level are seen only in the “dynamic” experiments (*Tap* and *Arm*) and are associated mainly with the SSF algorithm. For the rest of the experiments, no significant differences were noted, which shows that the AMPD, SSF, and PDA algorithms can be used interchangeably and do not significantly affect the resulting PRV for non-complex lab measurements. It is worth adding that the statistical tests marked in Table 2 indicate a smaller number of significant differences between the methods than for the subjects without outliers presented in Table 3.

An important element of this work was the verification of the quality of PRVs extracted from PPG in comparison with reference HRVs derived from ECG. This comparison is problematic as the two original waveforms often have different lengths due to artifacts affecting the PPG analyzing algorithms. This means that it is difficult to apply benchmarks that require the same amount of data to be compared, such as Pearson’s correlation, which is very often used in research but usually ignored when assessing the raw PRVs and HRVs, or the signals are locally averaged over time to HRs and only then compared. The approaches used in this study that are known from the literature, i.e., DTW, allowed comparing two fully corresponding signals represented, however, by a different number of samples. Spline interpolation resulted in uniform sampling, which successfully solved this problem.

### 4.4. Limitations of This Work

However, this study also has some limitations. The proposed methodology, focusing on using only the PPG signal, resolved the problem of deriving PRV in the presence of artifacts only to a certain extent. It should be taken into account that for measurements in everyday life scenarios, artifacts in PPG affecting the correctness of PRV determination will be visible even during the spontaneous arm or body movement, and these will increase in the case of more intense activities. These effects have been well illustrated in experiments with regular tapping on the smartwatch or arm movements (Table 2 and Table 3). In everyday life, random taps are rather rare but can introduce false peaks into the PPG signal, disrupting PRV locally. On the contrary, frequent movements of an arm or body are typical of activities during the day; therefore, problems with PRV derivation similar to those found in the *Arm* experiments are to be expected. It is different at night, when body movements are infrequent and happen irregularly.

The fact that the reduction of artifacts was restricted to PPG processing only, without the use of other sensors such as an accelerometer or gyroscope, resulted in the abandonment of the analysis of signals from the treadmill or everyday life, even though they were recorded at the experimental stage. Nevertheless, in this study, it was possible to check how much the quality of PRV can be improved without the use of additional signals, and thus, it will be useful in moving to the second stage of research on the currently omitted recordings.

There are also some shortcomings related to the selection of hyper-parameter values, which could be conducted more precisely with a greater amount of work, as well as with the selection of wavelets in the DWT/IDWT method, where it is worth carefully checking which of them have the greatest impact on improving the results.

Regarding the dataset, the signals collected from the Empatica E4 and during the treadmill and everyday life sessions were not analyzed in this article for reasons explained earlier. However, it is worth emphasizing here that those rich in content data are to be used in the next stage of our research for the additional optimization of some signal processing procedures, especially after supplementing them with methods that also employ ACC signals. Another limitation concerns the number of participants, which was just 11, including mostly male subjects of a similar age. Future experiments should involve more volunteers with greater diversity. The above facts prove that although the current solution improves the quality of PRV, it is still not quite sufficient for application in everyday life scenarios.

## 5. Conclusions

The main contribution of this article is to show the effective chain of procedures for processing the PPG signal from a smartwatch to eliminate artifacts without the use of any additional sensors, e.g., an accelerometer. The PPG signal is usually of poor quality, which is mainly due to the susceptibility to motion artifacts. Therefore, their reduction allows improving the quality of the extracted PRV with attention to its frequency content, which is crucial in its further analyses. Discrete wavelet transform and its inverse, followed by amplitude demodulation, enable the proper preparation of the PPG signal for the use of PRV extraction algorithms. In the tests, the SSF gave statistically more errors than the PDA or AMPD algorithms. It was found that the correction of outliers based on local statistical measures is only important when PPG is of low quality, and it has no effect under good signal quality. The same conclusion applies to post-processing using the AR model. The biggest problem turned out to be the interference from tapping the smartwatch and vigorous hand movements, which introduced additional peaks and modulation of the PPG amplitude. This suggests that it would be beneficial to use other information to further reduce such artifacts. For example, accelerometer signals can be utilized for that purpose. Nevertheless, our proposed procedures coped well with the artifacts in the remaining cases examined. Summarizing, the accuracy measures gathered in Table 2 and Table 3 (and especially the *r* coefficient) indicate that PRV derived solely from the PPG without the use of other sensors appears to be a reasonable alternative to HRV at rest, but it is unreliable, even after post-processing, during exercise.

Future work will mainly focus on the further studies on the impact of the PPG amplitude demodulation methods on the quality of the derived PRV. Additionally, the application of appropriate methods (including the nonlinear ones) exploiting the accelerometer signal will be investigated. This should allow for the efficient analysis of the results also from the treadmill and then from the everyday life records divided into nighttime and daytime activities.

## Figures and Tables

**Figure 1 sensors-22-07047-f001:**
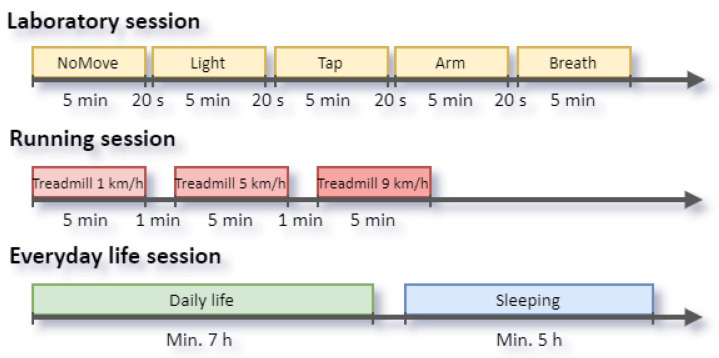
Experimental protocols for data collection. Only laboratory sessions were deeply analyzed in this paper.

**Figure 2 sensors-22-07047-f002:**
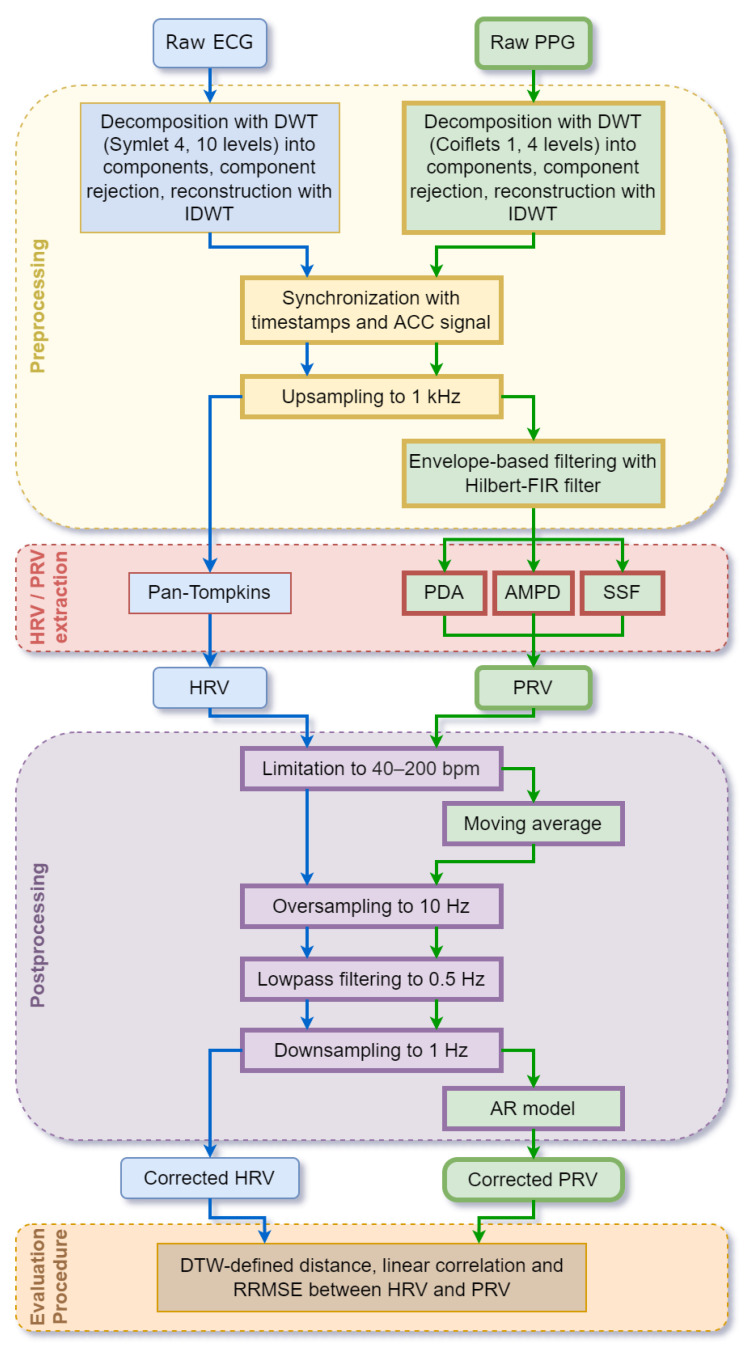
ECG and PPG processing flows. ECG is used as a reference to validate our new PPG processing method (blocks with thicker frames).

**Figure 3 sensors-22-07047-f003:**
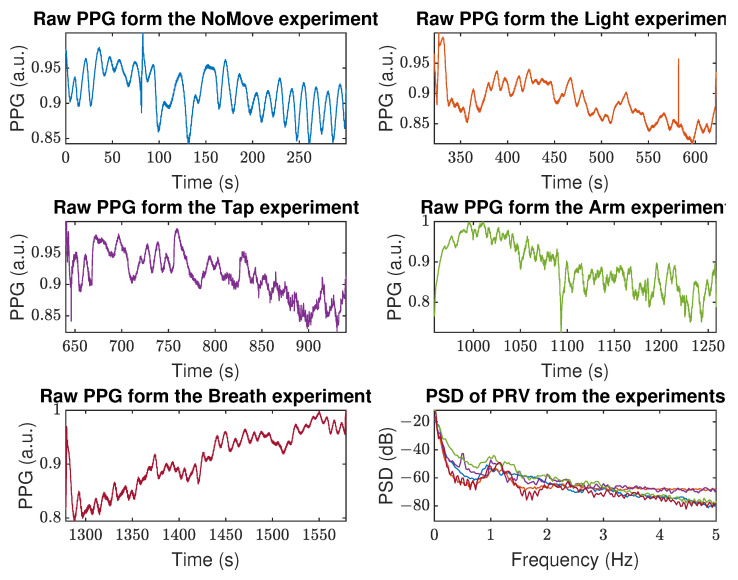
Raw PPG signals collected by the Samsung Watch 3 during the analyzed experiments together with their power spectral densities (PSD, lower-right panel), person P6. Colors of PPG signals are corresponding to their calculated PSDs.

**Figure 4 sensors-22-07047-f004:**
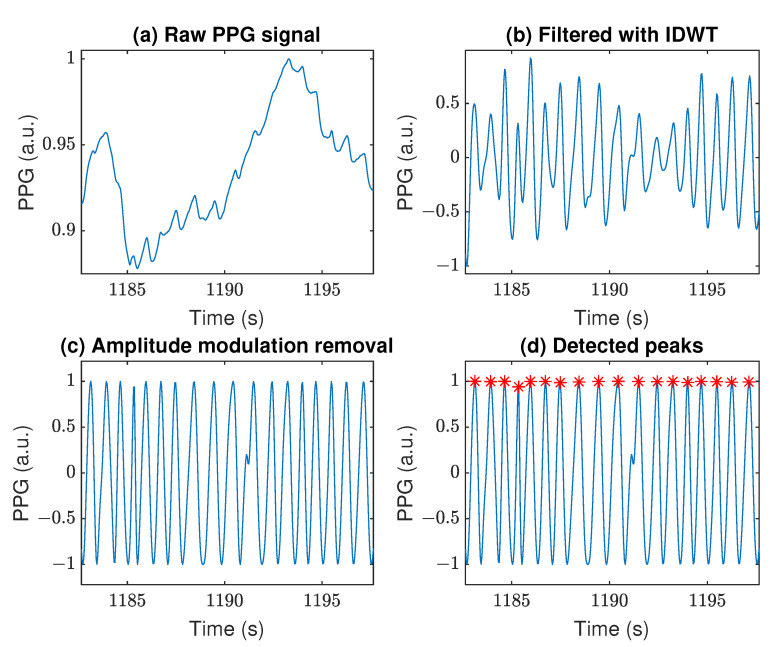
Example of PPG signal filtering: (**a**) Raw PPG signal obtained from the Samsung device, (**b**) Signal filtered with the IDWT, (**c**) Results of amplitude modulation removal, (**d**) Example of detected peaks (red asterisks) with the AMPD algorithm. The *Arm* experiment, person P6.

**Figure 5 sensors-22-07047-f005:**
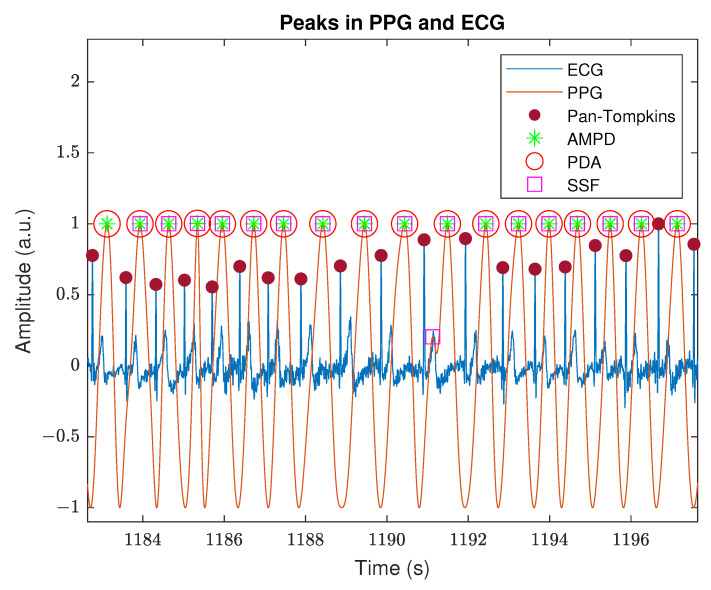
Visualization of peak detection in ECG and PPG signals. The *Arm* experiment, person P6.

**Figure 6 sensors-22-07047-f006:**
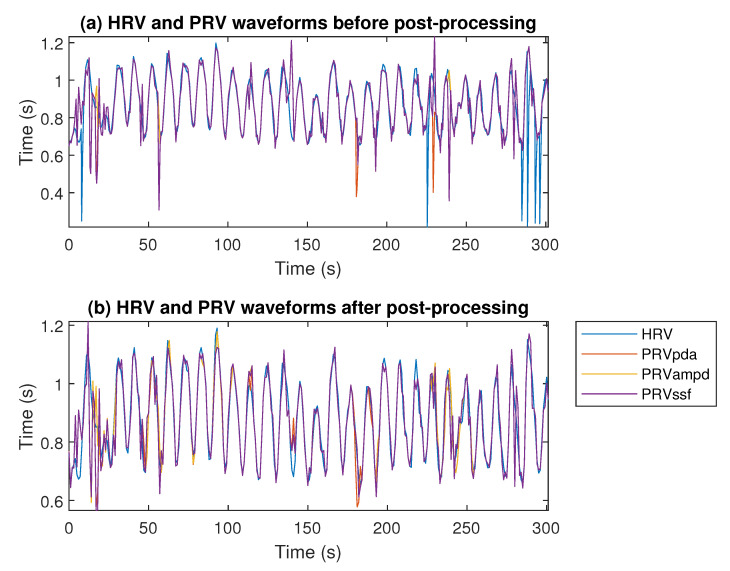
Comparison of the HRV and PRV waveforms for each algorithm before (**a**) and after (**b**) post-processing. The *Arm* experiment, person P6.

**Figure 7 sensors-22-07047-f007:**
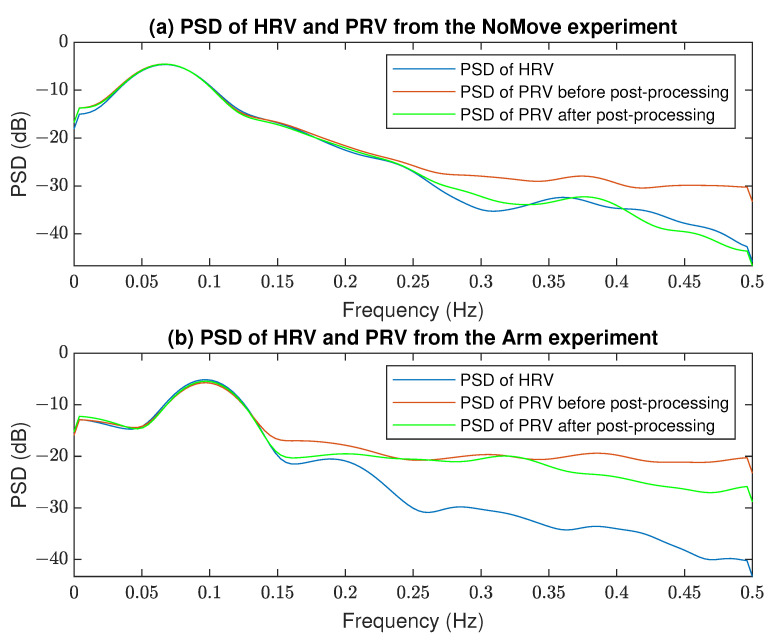
Comparison of the PSD of HRV and PRV before and after post-processing from the (**a**) *NoMove* and (**b**) *Arm* experiments, person P6.

**Table 1 sensors-22-07047-t001:** Measurement protocol with the activity clusters, task descriptions and task durations.

Task	Description	Duration [min]
*NoMove*	Sitting freely on a chair with hands resting on the table	5
*Light*	Changing the main room lighting every 10 s, participants position as before	5
*Tap*	Repeated finger tapping a Samsung in relation to the metronome rhythm	5
*Arm*	Raise arm to a vertical position above yourself (10 s) and lower it parallel to the desk (10 s)	5
*Breath*	Controlled very slow/fast breathing, 5/30 bpm determined by the displayed sine wave	5
*Walk*	Treadmill speed 1 km/h	5
*March*	Treadmill speed 5 km/h	5
*Run*	Treadmill speed 9 km/h	5
*Daily life*	Device worn during the day, various activities	≈420
*Sleeping*	Device worn before going to bed, overnight, monitoring participant sleep	≈420

**Table 2 sensors-22-07047-t002:** Similarity measures of HRVs and PRVs obtained from simultaneous ECG and PPG (averaged over ALL subjects): RRMSE—relative RMSE before AR modeling, RRMSEar—relative RMSE after AR modeling, DTWp—Dynamic Time Warping before entire post-processing, DTWs—Dynamic Time Warping after outliers correction, *r*—Pearson’s correlation coefficient after entire post-processing. The symbols #, *, or ¶ indicate that a given value is statistically smaller than for PDA, AMPD, and SSF, respectively, verified with the Student’s *t*-test at the 5% significance level.

Measure	PRV Algorithm	Experiments: ALL Subjects (Mean ± SD)				
		*NoMove*	*Light*	*Tap*	*Arm*	*Breath*
RRMSE (%)	PDA	**2.9 ± 1.6**	2.6 ± 1.7	22.5 ± 11.2	**17.7 ± 10.3**	24.4 ± 22.0
	AMPD	3.4 ± 2.1	**2.5 ± 1.5**	23.4 ± 12.8	20.0 ± 13.3	26.9 ± 23.7
	SSF	3.6 ± 2.2	2.7 ± 1.8	**22.9 ± 9.6**	18.4 ± 10.4	**23.7 ± 21.2**
RRMSEar (%)	PDA	**2.9 ± 1.0**	3.3 ± 1.5	21.5 ± 11.3	15.9 ± 9.5	23.4 ± 21.1
	AMPD	3.1 ± 1.3	**3.1 ± 1.4**	22.1 ± 12.6	18.9 ± 12.9	26.2 ± 23.0
	SSF	3.3 ± 1.3	**3.1 ± 1.4**	**21.5 ± 9.8**	**16.0 ± 9.2**	**22.9 ± 20.6**
DTWp	PDA	3.8 ± 2.1	4.5 ± 2.6	**38.9 ± 14.8**	32.5 ± 18.3	**28.2 ± 21.8**
	AMPD	**3.5 ± 1.6**	**4.5 ± 2.5**	37.5 ± 19.2	32.7 ± 18.8	31.0 ± 24.3
	SSF	4.7 ± 3.1	4.9 ± 3.0	42.2 ± 13.3	**32.3 ± 17.5**	29.4 ± 22.9
DTWs	PDA	**3.3 ± 1.3**	**3.7 ± 1.5**	36.2 ± 15.8	**24.8 ± 12.9**	23.7 ± 18.2
	AMPD	3.6 ± 1.6	**3.7 ± 1.5**	34.8 ± 17.7	25.4 ± 14.0	24.5 ± 18.7
	SSF	3.9 ± 2.0	4.1 ± 2.0	**37.5 ± 13.0**	25.8 ± 13.4	**23.4 ± 17.9**
*r*	PDA	0.96 ± 0.05	**0.95 ± 0.06**	0.66 ± 0.16	0.75 ± 0.15 *^,¶^	**0.87 ± 0.15**
	AMPD	0.96 ± 0.05	0.96 ± 0.05	**0.67 ± 0.14**	0.77 ± 0.13	0.87 ± 0.18
	SSF	**0.95 ± 0.05**	0.95 ± 0.07	0.65 ± 0.16	**0.74 ± 0.15** *^,#^	0.85 ± 0.19

**Table 3 sensors-22-07047-t003:** Similarity measures of HRVs and PRVs obtained from simultaneous ECG and PPG (averaged over SELECTED subjects): RRMSE—relative RMSE before AR modeling, RRMSEar—relative RMSE after AR modeling, DTWp—Dynamic Time Warping before entire post-processing, DTWs—Dynamic Time Warping after outliers correction, *r*—Pearson’s correlation coefficient after entire post-processing. The symbols #, *, or ¶ indicate that a given value is statistically smaller than for PDA, AMPD, and SSF, respectively, verified with the Student’s *t*-test at the 5% significance level.

Measure	PRV Algorithm	Experiments: SELECTED Subjects (Mean ± SD)				
		*NoMove*	*Light*	*Tap*	*Arm*	*Breath*
RRMSE (%)	PDA	**2.9 ± 1.6**	2.6 ± 1.7	13.8 ± 5.6 ^¶^	11.1 ± 4.2	**4.8 ± 3.8**
	AMPD	3.4 ± 2.1	**2.5 ± 1.5**	**13.1 ± 6.1** ^¶^	**10.7 ± 3.9** ^¶^	9.2 ± 8.3
	SSF	3.6 ± 2.2	2.7 ± 1.8	16.1 ± 4.8	11.9 ± 4.1	5.4 ± 4.2
RRMSEar (%)	PDA	**2.9 ± 1.0**	3.3 ± 1.5	12.3 ± 4.9 ^¶^	9.8 ± 3.8	3.9 ± 2.4
	AMPD	3.1 ± 1.3	**3.1 ± 1.4**	**11.7 ± 5.4**	**9.7 ± 3.8**	7.5 ± 6.3
	SSF	3.3 ± 1.3	**3.1 ± 1.4**	14.3 ± 4.2	10.3 ± 4.0	**3.9 ± 2.0**
DTWp	PDA	3.8 ± 2.1	4.5 ± 2.6	30.5 ± 11.2 ^¶^	21.7 ± 8.6	**12.2 ± 7.6**
	AMPD	**3.5 ± 1.6**	**4.5 ± 2.5**	**23.0 ± 8.8** ^¶^	**21.4 ± 8.7**	13.1 ± 8.7
	SSF	4.7 ± 3.1	4.9 ± 3.0	37.4 ± 13.6	22.4 ± 9.2	12.5 ± 7.9
DTWs	PDA	**3.3 ± 1.3**	**3.7 ± 1.5**	25.4 ± 9.4	17.4 ± 6.2 ^¶^	10.5 ± 6.3
	AMPD	3.6 ± 1.6	**3.7 ± 1.5**	**21.3 ± 8.3** ^#,¶^	**16.9 ± 6.0** ^¶^	11.0 ± 6.8
	SSF	3.9 ± 2.0	4.1 ± 2.0	30.9 ± 11.2	18.1 ± 6.5	**10.4 ± 6.1**
*r*	PDA	0.96 ± 0.05	**0.95 ± 0.06**	0.64 ± 0.2	0.78 ± 0.15	0.92 ± 0.11
	AMPD	0.96 ± 0.05	0.96 ± 0.05	0.66 ± 0.16	0.79 ± 0.14	**0.92 ± 0.09**
	SSF	**0.95 ± 0.05**	0.95 ± 0.07	**0.63 ± 0.19**	**0.77 ± 0.14** *	0.91 ± 0.12

## Data Availability

The data presented in this study are available on reasonable request from the corresponding author. The data are not publicly available due to ethical constraints.

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
