# Peer review of "Processing Photoplethysmograms Recorded by Smartwatches to Improve the Quality of Derived Pulse Rate Variability"

_sensors, 2022, doi:10.3390/s22187047_

Round 1

Reviewer 1 Report

this paper aims to solve the signal problem of using wearable photoplethysmography (PPG) for cardiac monitoring. Two commercially available smartwatches were used, one for PPG signal collection and the other for reference of using ECG. 

major comments are as follows:

1. why 1185 to 1195 seconds are especially interested? how is the data or signal within a procession of 5 min for some kind of activity?

2. how is the spectrum of PPG and ECG data for some kind of activity?

3. which activity is recorded and presented in Fig 3,4,5?

4. every activity signal should be presented in figures to show the visual difference among them。

5. in line 487, focused should be focusing; and, in line 493, resulted should be resulting. please also check all other similar grammar problems.

Reviewer 2 Report

Review - Processing Photoplethysmograms Recorded by Smartwatches to Improve the Quality of Derived Pulse Rate Variability

 This paper present and discuss the way to obtain PRV data based on the artifacts reduction for PPG signals processing from two smartwatches. Overall, the paper is well-written and organized.  The state-of-the-art review, methods, analysis, results discussion, and references are pretty good.  Peak detection is performed with several algorithms and RMSE and t-Student tests are applied to evaluate the algorithms performance.

Please consider the following comments and recommendations:

- In Table 3 only “SELECTED subjects” are represented. This table is redundant since the results from all data are presented in Table 2.

- In the paper, the authors state that “we performed a statistical paired t-Student test with the 5% significance level to check for differences in the performance of the PRV extraction algorithms”. However, it is not clear if the t-Student was performed between the HRV and PRV data or between PRV data from different algorithms.

- The authors state that “This is mainly because the strong tapping introduced notably erroneous peaks into the PPG signal.” It would be interesting to evaluate the peak detection accuracy difference between the NoMove and Tap sections. 

Reviewer 3 Report

An interesting study on the ability of PPG to track HRV during different conditions.

Abstract – what is “AR”

Intro –

“which high diagnostic ability has been demonstrated in recent years” – rephrase

“Nevertheless, more complete approaches to directly comparing HRV and PRV signals are still extremely needed” – remove extremely

“The main contribution of this study relays on proposing…”  relays is not an appropriate term for this

Materials and Methods-

Thank you for providing sampling rate for sensor data.

What software was used to record the Polar H10 waveform?

Justification of the methods-

You state that the H10 has been validated with respect to RR intervals.  However, this was always done using the RR output from the module itself.  We have no published data indicating what the 130 Hz waveform is capable of.  The 130 Hz is what the API allows, but it’s possible that the internal sample rate is higher and not upsampled.  Please consider this point and adjust the conclusion.

Discussion-

Please comment on the effects of upsampling the low native sample rates of both the ECG and PPG devices on HRV precision.  Most (if not all) recommendations for device sample rates are a minimum of 200-250 Hz (Kwon, Ohhwan, et al. "Electrocardiogram sampling frequency range acceptable for heart rate variability analysis." Healthcare informatics research 24.3 (2018): 198-206. Shaffer F and Ginsberg JP (2017) An Overview of Heart Rate Variability Metrics and Norms. Front. Public Health 5:258. doi: 10.3389/fpubh.2017.00258).

Further comment on the effects on non-regular tapping/motion effects on the potential resulting HRV distortions should be made.

Comment on the potential effect of reduced limb blood flow during high intensity exercise on the PPG quality/HRV assessment.

In my opinion a major conclusion of this study is that even with post processing methods, HRV derived from PPG is unreliable during exercise but on the other hand appears to be a reasonable alternative to ECG while at rest (i.e., sleeping).  Please comment on this (with reference to r values) in the conclusion.

Reviewer 4 Report

The article is very well constructed and presented in a sequential manner as needed considering the authors have used MATLAB as the tool for analyzing the signals and processing them. There are a few major issues regarding the research methodology presented here:

1)     Why is the data from Empatica E4 not considered in this study? Isn’t it going to be reference data when compared to Samsung Galaxy Smartwatch3 data? Only 2 devices were used in this study. Unless multiple devices (n>2) of the same types are studied simultaneously and data are presented from all, there cannot be a clear conclusion with negligible bias.

2)     Why is the data from the running session and everyday life session not considered? These two sessions are the ones that could have added more variability because of continuous body movements. The objective of the study is to improve pulse rate variability and how a trained laboratory session with the least amount of variability will secure the objective of this study?

3)     The accelerometer is an important feature of every smartwatch and the data from the accelerometer is omitted from the study and not even correlated to the PRV and ECG data since running and everyday life sessions were taken out of the study. The major causes of PRV due to random as well as rhythmic body movements are not studied in this work.

This might be a model study but it is not a complete study that can point out the relevance and uniqueness of this work presented. I appreciate the authors’ effort to completely base the study on PPG recordings and improve them but in reality, it is impossible to improve a device’s accuracy in removing PRV using a single sensor’s data because it is not considered that efficient to date.

Round 2

Reviewer 1 Report

all comments are well addressed.

Reviewer 3 Report

Great job with the clarifications/corrections.  I look forward to citing this study when it's published.

Reviewer 4 Report

Considering an initial approach towards a single sensor data related improvements, I appreciate the authors' effort. The justifications mentioned for the review comments are enough for the time being.